# Spin-Peierls, Spin-Ladder and Kondo Coupling in Weakly Localized Quasi-1D Molecular Systems: An Overview

**Jean-Paul Pouget**

Laboratoire de Physique des Solides, Université Paris-Saclay, CNRS, 91405 Orsay, France;
jean-paul.pouget@u-psud.fr

**Abstract:** We review the magneto-structural properties of electron–electron correlated quasi-one-dimensional (1D) molecular organics. These weakly localized quarter-filled metallic-like systems with pronounced spin 1/2 antiferromagnetic (AF) interactions in stack direction exhibit a spin charge decoupling where magnetoelastic coupling picks up spin 1/2 to pair into S = 0 singlet dimers. This is well illustrated by the observation of a spin-Peierls (SP) instability in the (TMTTF)2X Fabre salts and related salts with the o-DMTTF donor. These instabilities are revealed by the formation of a pseudo-gap in the spin degrees of freedom triggered by the development of SP structural correlations. The divergence of these 1D fluctuations, together with the interchain coupling, drive a 3D-SP ground state. More surprisingly, we show that the Per2-M(mnt)2 system, undergoing a Kondo coupling between the metallic Per stack and the dithiolate stack of localized AF coupled spin $\frac{1}{2}$ (for M = Pd, Ni, Pt), enhances the SP instability. Then, we consider the zig-zag spin ladder DTTTF2-M(mnt)2 system, where unusual singlet ground state properties are due to a combination of a $4k_F$ charge localization effect in stack direction and a $2k_F$ SP instability along the zig-zag ladder. Finally, we consider some specific features of correlated 1D systems concerning the coexistence of symmetrically different $4k_F$ BOW and $4k_F$ CDW orders in quarter-filled organics, and the nucleation of solitons in perturbed SP systems.

**Keywords:** 1D organic conductor; electron–electron correlated system; spin-charge decoupling; magnetoelastic coupling; spin-Peierls transition; kondo coupling; spin ladder

## 1. Basic Interactions in Quasi-1D Molecular Systems

Since the discovery of a metal-like conductivity in 1954, when perylene (Per) was exposed to Br, a new field of research opened among organic materials, with the synthesis in 1960 of the molecular acceptor TCNQ (tetra-cyanoquinodimethane), and in 1970 the discovery of the molecular donor TTF (tetra-thiafulvalene), rapidly followed by the combination of acceptor (A) and donor (D) stacks in the same structure. This led to a large family of charge-transfer organic conductors such as Qn(TCNQ)2 in 1960, NMP-TCNQ in 1965, and then TTF-TCNQ in 1972 (the chemical name of organic molecules quoted in the text is given in the annex). These findings were followed by the unexpected discovery in 1977 that, when exposed to AsF5, the conductivity of trans-polyacetylene, (CH)x, can be raised by more than seven orders of magnitude. These pioneering works open an incredibly large and fecund area of research, bolstered by the continuous discovery of new physical phenomena, now forming the so-called domain of "organic electronics". These finding were rapidly followed by potential applications and the realization of technological devices [1].

The originality of organic metals is due to the fact that these materials exhibit low 1D or 2D electronic properties when coupled to a soft molecular structure. The low electronic dimension generally triggers electronic (charge or spin) collective density wave (CDW or SDW) instabilities not found in conventional 3D metals. In addition, because of the presence of a soft underlying lattice, the electronic instability allows the stabilization of unconventional ground states involving coupled charge/spin and lattice modulations, the

most popular one being the $2k_F$ Peierls ground state found in 1D systems such as trans-polyacetylene. Incommensurate $2k_F$ CDW/Peierls modulations were initially detected in 1D metallic charge transfer salts such as $TTF^{+\rho}$-$TCNQ^{-\rho}$, formed of segregated stacks of donor (D) and acceptor (A) with a D to A charge transfer of $\rho = 0.59$, and where the Fermi wave vector of each individual 1D electron gas is simply given by $k_F = \rho/4$ in a reciprocal chain unit. CDW physics, conjointly observed in 1D inorganic systems, have been recently summarized in [2]. Another remarkable originality of organic quasi-1D systems is that, due to the large spatial extent of organic molecules, the intra-stack electronic transfer integral, $t_{//}$, is comparable to the intra-site (U) and first neighbor inter-site ($V_1$, $V_2$ ... ) Coulomb repulsions entering into the 1D extended Hubbard model, which models the properties of the correlated 1D electron gas quite well [3]. This allows the rationalization of the experimental observation of various $2k_F$ density wave instabilities, as well, to account for the unexpected CDW instability (discovered in TTF-TCNQ) at the $4k_F$ wave vector, where the $4k_F$ wave vector is the first Fourier component of a lattice of localized charge of the Wigner type (for more detail, see [4]). Although $4k_F$ collective charge localization waves were initially observed in incommensurately filled electronic systems, a stronger effect occurs in salts with one charge per site or every two sites. Thus, special attention has been devoted in the literature to studying half-filled ($\rho = 1$) and quarter-filled ($\rho = 1/2$) organic systems, leading to a better interplay between charge and antiferromagnetically (AF) coupled spin degrees of freedom. These commensurate systems allow the efficient stabilization of ground states not known in 3D materials, such as the spin-Peierls (SP) ground state typical of 1D systems [4]. SP ground states stabilized in many of them are more specifically considered in this review.

The basic aspect of the electronic phase diagram of half-filled and quarter filled 1D electron gas is described in refs. [5–9], respectively. Then, the modification of phase diagrams, taking explicitly into account the important coupling of electrons with lattice degrees of freedom (electron–phonon and magnetoelastic coupling), was considered in refs. [10,11]. Figures 1 and 2 exhibit a large panel of instabilities and of ground states experimentally observed in various half-filled and quarter-filled 1D molecular systems, respectively. In particular, both Mott insulator and 1D Luttinger liquids stabilized by sizeable electron–electron correlations present either antiferromagnetic or spin-Peierls ground states (Figures 1 and 2). There are, however, subtle differences between half-filled and quarter-filled situations which depend on the degree of charge localization and electronic interactions, such as the magnitude of the gap of charge, $\Delta\rho$, an the antiferromagnetic exchange interaction J between $S$-$\frac{1}{2}$ (Sections 2 and 3).

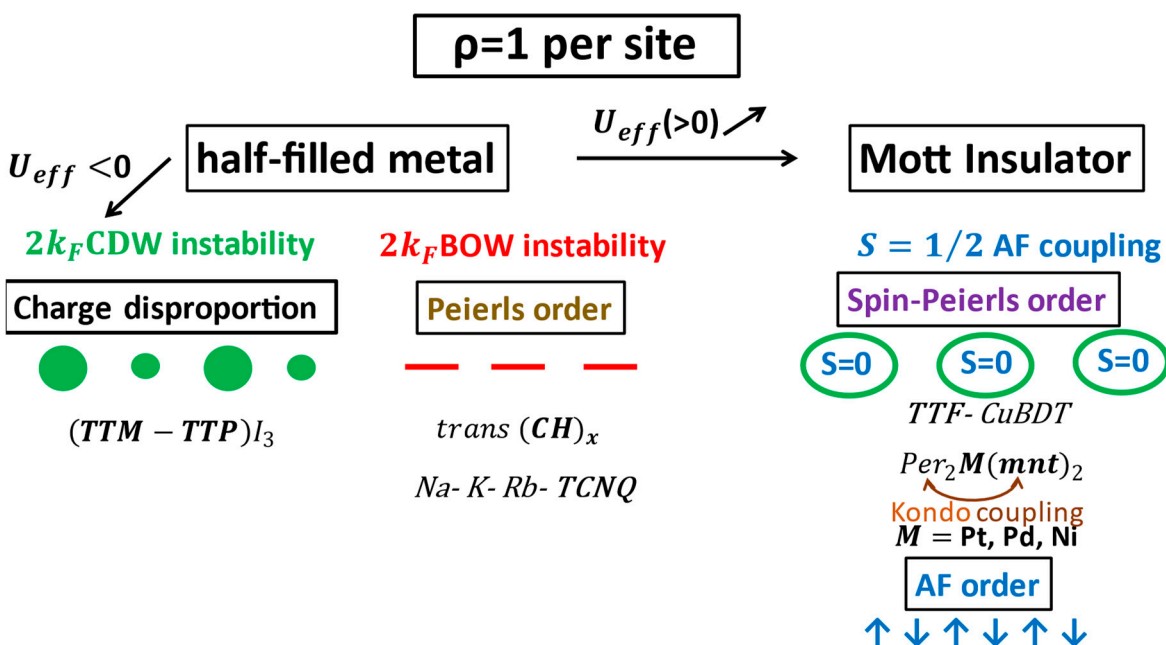

**Figure 1.** Electronic phase diagram of the half-filled chain. The various electronic instabilities and ground states are schematically described. Typical experimental examples are also indicated. The stack bearing the instability is in bold text.

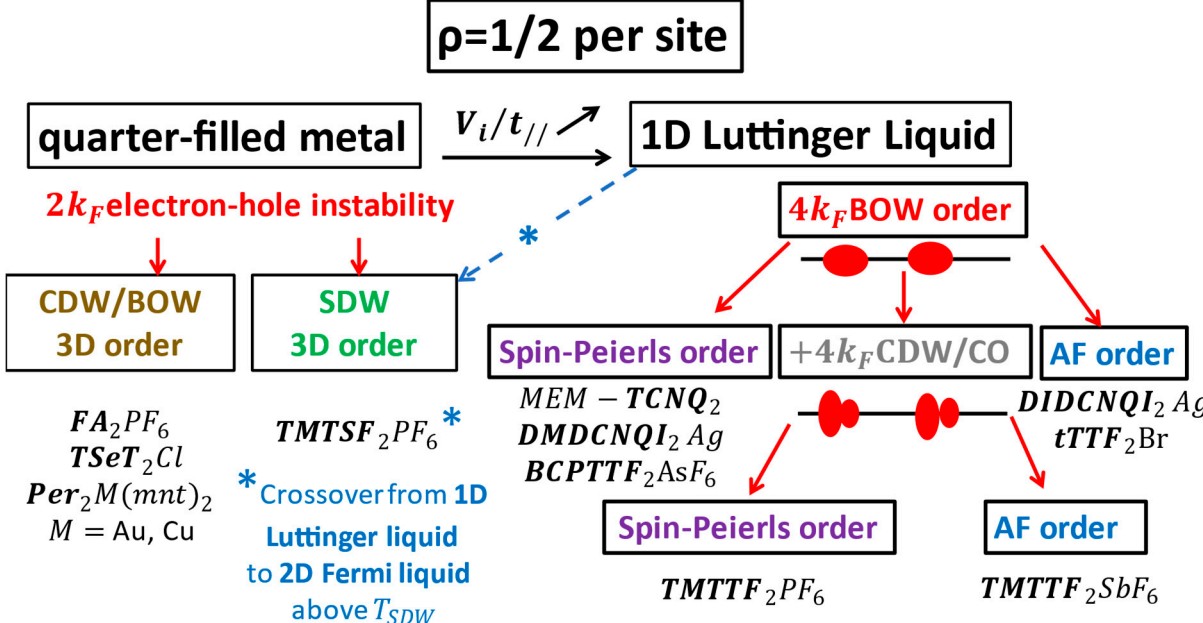

**Figure 2.** Electronic phase diagram of the quarter-filled chain. The various electronic instabilities and ground states are schematically described. Typical experimental examples are also indicated. The stack bearing the instability is in bold text. The crossover from a 1D Luttinger liquid to a 2D Fermi liquid in TMTSF$_2$PF$_6$ is indicated by the symbol * in the figure.

## 2. Half-Filled ($\rho = 1$) Versus Quarter-Filled ($\rho = 1/2$) Organic Systems

The dilemma particularly concerns quarter-filled 1D dimerized systems of current interest whose electronic structure is basically that of a quarter-filled HOMO (LUMO) band for the donor (acceptor) stack ($\rho = 1/2$). However, due to lattice dimerization, a band gap, $2\Delta_D$, opens at the Brillouin zone boundary $\pm\pi/a$, where a is the chain parameter of the dimerized stack. For non-interacting electron, this splits the HOMO band by

twice the difference of intra- and inter-dimer transfer integrals: $2\Delta_D = 2 | t_{intra} - t_{inter} |$. Thus, if only the upper band of the dimerized chain is relevant, the electronic system could be considered as half-filled ($\rho = 1$). If not, it should be considered as quarter-filled ($\rho = 1/2$). The clarification of the dilemma requires a consideration of all the interactions, knowing that lattice dimerization (or $4k_F$ BOW ordering of the stack- see Figure 2) induces a $4k_F$ lattice potential, allowing second-order electron–electron Umklapp scattering which, in the presence of large electron–electron repulsions, tends to localize the charges [12,13]. By taking into account these effects, the gap of charge, $\Delta\rho$, relevant for the charge localization process, becomes a complex function of $\Delta_D$ but also of the coulomb interaction parameters [14].

Thus, if in the T range of interest (let us say below 300 K) $\Delta\rho$ is larger than 1000 K($\sim\pi T_{max}$), the system can be considered as effectively half-filled. In this limit, each electron is localized in the bonding state (antibonding state) of the acceptor (donor) dimer. Charges are frozen, so that there is no charge degree of freedom available in the dimer. This means that there is no possibility for a charge disproportion (or charge ordering, CO) inside the dimer. As in the case for a half-filled system (Figure 1), the only low T instabilities are those driven by Heisenberg AF coupling between localized S-$\frac{1}{2}$ in each dimer, eventually coupled to a phonon field in order to achieve an SP ground state. This situation occurs in quarter-filled strongly localized $A_2Y$ compounds, where Y is a monovalent cation, such as MEM(TCNQ)$_2$, which exhibits at 335K a first order transition to a $4k_F$ BOW phase (with dimerization of the acceptor TCNQ stack accompanied by a modification of the cation Y sublattice), opening a very large energy gap of $\Delta\rho = 0.64$ eV, then an SP ground state at 18 K. To a lesser extent, this also occurs in quarter-filled $D_2X$ salts, where D is a derivative of the TTF molecule and X a monovalent anion, adopting the prototypal structure of the (TMTSF)$_2$X/(TMTTF)$_2$X (i.e., Bechgaard/Fabre salts) [15]. In this category, organics (DIMET)$_2$SbF$_6$ and (t-TTF)$_2$Br (where $\Delta\rho \sim 3400$ K) exhibit an AF ground state at 12 K and 35 K, respectively, while (BCP-TTF)$_2$PF$_6$ and AsF$_6$ (where $\Delta\rho \sim 2000$ K) exhibit an SP ground state at 32.5 K and 36 K, respectively.

In the opposite situation, where the gap of charge is smaller, $\Delta\rho \sim \pi T_\rho < 1000$ K, the system is really quarter-filled with non-frozen intra-dimer charge degrees of freedom. In these systems, $T_\rho$ is the temperature of minimum of conductivity, corresponding to the $4k_F$ BOW charge localization temperature. This situation is relevant for the Fabre salts (TMTTF)$_2$X where $T_\rho \sim 230$ K, with an activation energy of conductivity of below $\frac{\Delta\rho}{2} \sim 350$ K. At intermediate temperature, around 100 K, those salts undergo a CO/ferroelectric phase transition causing a polar charge rearrangement inside the dimers (Figure 2), accompanied by an anion X displacement. This transition is followed at lower temperature by another phase transition to either AF or SP ground states (for a recent review on quarter-filled organic systems, see [16]). Note that, for a true quarter band-filling, fourth-order Umklapp scattering processes are relevant to interpret the physical properties, as in the Fabre salts [13].

## 3. Antiferromagnetic Coupling in Spin-1/2 Weakly Localized Organic Stacks

When charges are localized, their S-$\frac{1}{2}$ interact via an antiferromagnetic Heisenberg coupling. In the case of first neighbor interaction, J, the Hamiltonian simply reads:

$$H_{spin} = J \sum_i S_i S_{i+1} \tag{1}$$

No magnetic transition occurs for an isolated S = 1/2 AF chain due to thermal and quantum fluctuations. In fact, in the presence of these fluctuations, the spin susceptibility, $\chi_{spin}$, can be exactly calculated [17]. From both the magnitude and the thermal dependance of $\chi_{spin}$, the first neighbor AF J interaction defined by (1) can be obtained. The fit of the experiment data of organics using expression (1) leads to a large panel of J values. J increases along the sequence J = 77 K for the half-filled ($\rho = 1$) TTF donor chain of the 1:1 TTF—CuBDT compound [CuBDT is CuS$_4$C$_4$(CH)$_4$] and, for various A and D quarter-filled

chain ($\rho$ = 1/2) 2:1 compounds, J = 138 K in MEM-(TCNQ)$_2$ [4], J = 270 K (see Figure 4 in Section 4) in (BCP-TTF)$_2$AsF$_6$ [18], J = 460 K in (TMTTF)$_2$PF$_6$-D$_{12}$ [19](see Figure 3), and similar J values are found for other Fabre salts with different anion X [20]; somewhat larger J are found for the substituted donor o-DMTTF: J = 490(30)K–530(30)K in (o-DMTTF)$_2$X with X = Br and I, respectively, [21] and J~520(50)K with X = NO$_3$ [22]. The increase of J seems to be correlated with an enhanced charge delocalization in the stack direction. Note that a satisfactory fit of J is obtained if the experimental spin susceptibility (measured at 1 bar) is corrected by the thermal volume expansion. This correction is shown in Figure 3 for deuterated $TMTTF_2PF_6$. In these Fabre salts, J is quite large, which means that the spins (charge) are weakly localized. Additionally, for a significantly delocalized spin system, $\chi_{spin}$, can be equally well accounted from a first neighbor 1D extended Hubbard model. Such a calculation, detailed in note [23], is also shown in Figure 3 which provides Coulomb interaction parameters U and V$_1$, scaled by the DFT transfer integral t $\approx$ 0.18–0.2 eV on the TMTTF stack. These values are twice as small as those calculated for the TTF stack of TTF-TCNQ [24]. Note that deviation between the fit and experimental data of Figure 3 could be reduced at high temperatures by using a larger U (~6t), and at low temperatures by including a non-negligeable second neighbor interaction V$_2 \approx$ V$_1$/2.

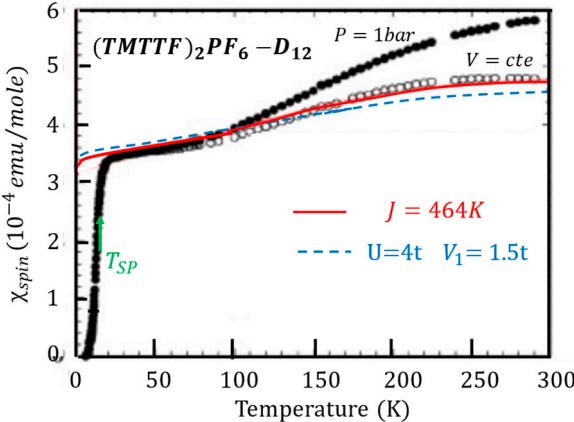

**Figure 3.** Temperature dependence of the spin susceptibility, $\chi_{spin}$, of deuterated $TMTTF_2PF_6$. The filled circles give the ambient pressure SQUID measurements, and the empty circles give the spin susceptibility at constant volume. The continuous red line is the fit of $\chi_{spin}$ with the expression of the spin susceptibility of the localized *S*—1/2 AF Heisenberg chain [17]. The adjustment shown in the figure, which takes into account both the absolute value and the thermal dependence of $\chi_{spin}$, gives *J* $\approx$ 464 K [19]. The dotted blue line is the fit of $\chi_{spin}$ for the 1D extended Hubbard model, as detailed in note [23].

A large range of first neighbor AF coupling J$_1$ values is also obtained from the fit of the high temperature spin susceptibility, $\chi_{spin}$, in the half-filled dithiolate chain ($\rho$ = 1) of $Per_2$M(mnt)$_2$ [25]: J$_1$ = 38 K for M = Pt, from 100 K to 240 K in M = Ni and 270 K for M = Pd (see Figure 7 in Section 5). $\chi_{spin}$ is additionally modified upon cooling, firstly by the SP fluctuations, and then by the Kondo interaction with the electronic degrees of freedom on the Per stack for M = Ni and Pd. Kondo interactions add a frustrated second neighbor AF coupling J$_2$ on the dithiolate stack (see Section 5).

## 4. Spin-Peierls Fluctuations on Donor Stacks in D$_2$X Weakly Localized Conductors

All AF organics considered in the last section are subject to a spin-Peierls (SP) instability when the S = 1/2 AF donor stack is coupled to a phonon field. This leads to a 3D phase transition at finite $T_{SP}$ toward an S = 0 non-magnetic ground state accompanied by a dimerization of the S = 1/2 AF chain. Additionally, this second order phase transition is announced by an important regime of SP pre-transitional structural fluctuations which develop below the so-called mean-field SP transition temperature $T_{SP}^{MF}$ of the chain, which

is significantly higher than $T_{SP}$. In most of the SP compounds, magneto-structural coupling picks up singlet fluctuations of the AF chain and correlates them into an SP short-range order. This local order is revealed by a decrease of $\chi_{\mathrm{spin}}$, due to the formation of a pseudo-gap in the spin degrees of freedom, which develops in the T range of structural fluctuations below $T_{SP}^{MF}$ [26]. The observation of a pseudo-gap in $\chi_{\mathrm{spin}}$ is the signature of a classical (adiabatic) regime of instability. It is found in many SP materials [2,4,18,27], some of which are considered below. However, there are other compounds such as TTF—CuBDT, MEM-(TCNQ)$_2$ [4,18] and Per$_2$Pt(mnt)$_2$ [25] where SP pre-transitional lattice effects do not affect the spin susceptibility. There, the SP instability occurs in the anti-adiabatic (quantum) regime of fluctuation. Anti-adiabatic effects are observed in materials exhibiting the smallest AF exchange interaction J, or where charge/spin are strongly localized. Non-adiabaticity is due to the fact that the phonon energy $\hbar\Omega_0$ of the critical lattice mode is larger than the magnetic energy ~J [28]. More precisely, the classical–quantum crossover for an SP transition in a Heisenberg chain occurs for $\hbar\Omega_0 \approx \Delta_{\mathrm{MF}}/2$, where the mean-field SP gap, $\Delta_{MF}$, is related to $T_{SP}^{MF}$ by $\Delta_{MF} \approx 2.47 k_B T_{SP}^{MF}$ [29]. Additionally, when non-adiabaticity effects are important, the Hamiltonian (1) is modified by the introduction of next near neighbor exchange interactions. This is the case for the SP inorganic compound CuGeO$_3$ [30].

In the adiabatic limit, SP fluctuations develop a pseudo-gap in the magnetic excitation spectrum of the Heisenberg chain. The energy dependence of the pseudo-gap is controlled by the spatial extension of the SP correlation length, quantitatively calculated in ref. [26]. Figure 4 shows the associated drop of spin susceptibility below about $T_{SP}^{MF} \approx 120$ K in (BCP-TTF)$_2$AsF$_6$. It occurs in the temperature range where quasi-1D SP fluctuations are developing (Figure 4). Below $T_{SP} = 35$ K, these short-range fluctuations condensed into 3D superlattice reflections, which implies a long-range dimerized SP order [18,26,31]. The development of the SP pseudo-gap in deuterated $TMTTF_2PF_6$ has been precisely studied by inelastic neutron scattering measurements of the energy dependance of magnetic excitations. It reveals a drop in the magnetic density of states in an energy range of $\Delta_{MF}$ upon cooling below $T_{SP}^{MF} \approx 40$ K [32]. Figure 5 shows both the drop in spin susceptibility and, in inset the decrease in amplitude of magnetic excitation energy. The pseudo-gap, a precursor at an SP transition, is also found in other D$_2$X weakly localized conductors with large J such as (o-DMTTF)$_2$NO$_3$ [22].

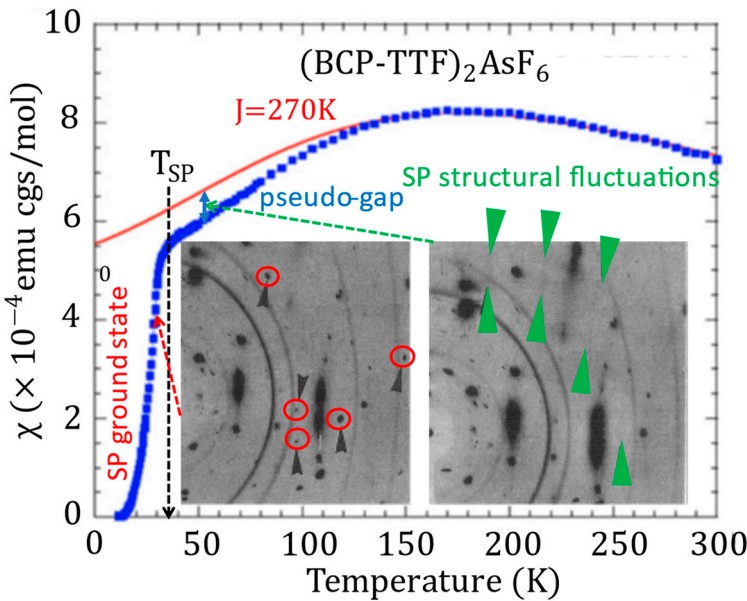

**Figure 4.** Spin susceptibility of $(BCP\text{-}TTF)_2AsF_6$, showing the development of a pseudo-gap below $T_{SP}^{MF} \approx 120$ K when quasi-1D SP fluctuations of chain dimerization develop. Spin susceptibility abruptly drops at the $T_{SP} = 35$ K SP phase transition, below which satellite reflections due to the dimerization are detected. The inset presents the $PF_6$ salt X-ray diffuse scattering patterns taken (right part) in the regime of quasi-1D SP fluctuations (green arrows), and (left part) in the 3D-SP dimerized phase (superlattice reflections surrounded by red circles). The figure combines results taken from refs. [18,31].

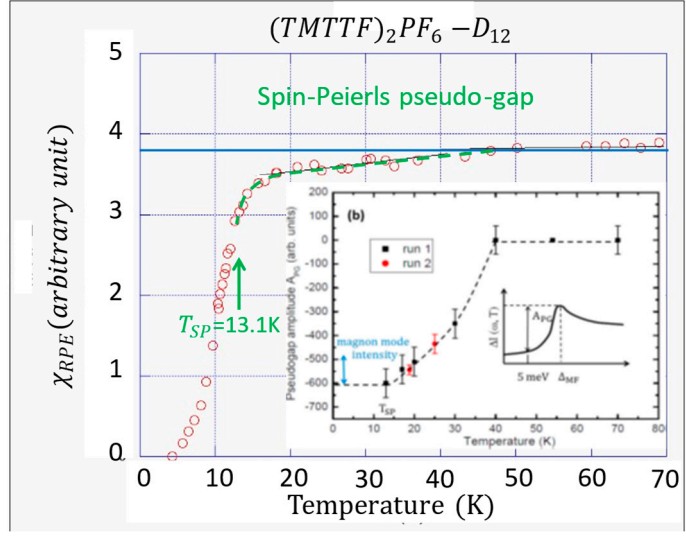

**Figure 5.** Spin susceptibility of deuterated $TMTTF_2PF_6$ near the SP transition, showing the development of a pseudo-gap below 40 K ($\chi_{RPE}$ has been kindly provided by C. Coulon). The inset gives the thermal dependance of the amplitude of the pseudo-gap at 5 meV in the same T range. The energy dependence of the pseudo-gap, obtained from neutron measurements [32], is also schematically represented.

## 5. Kondo Coupling between Localized and Delocalized Stacks in the Per$_2$-M(mnt)$_2$ Series

A very original family of 2:1 charge transfer salt is the $\alpha$-Per$_2$-M(mnt)$_2$ series, whose array projected perpendicularly onto the chain direction is shown in Figure 6a. It contains a metallic quarter-filled Per stack subject to a Peierls instability and dithiolate chains. For M = Au and Cu close shell dithiolate molecules are non-magnetic, while for M = Pt, Ni

and Pd dithiolate derivatives, localized S = 1/2 chains with AF interactions are formed (Figure 6b). Such a coupled stack array presents very original physical properties which are reviewed in [1,33]. It exhibits, in particular, three types of substantial interchain exchange interactions $J_\perp$ (named J′, J″ and J‴ in Figure 6a), whose importance is assessed by the observation of a single EPR line at a g value intermediate between those of dithiolate and Per molecules in the Pd [34] and Pt [35] derivatives. This coupling allows a subtle interplay between Peierls and spin-Peierls instabilities, respectively, located in the Per and M = Pt, Ni and Pd dithiolate stacks [36]. In this respect, and based on the finding of substantial $J_\perp$ values, a Kondo-type of inter-chain magnetic coupling (Figure 6b) was proposed to be a relevant interaction in the Pt salt [35]. However, it was recently realized [25] that most visible effects arise from Per-dithiolate Kondo coupling in the less studied Ni and Pd salts. In them, the SP instability occurs in the adiabatic regime, at the difference of the non-adiabatic SP regime of the Pt salt [25].

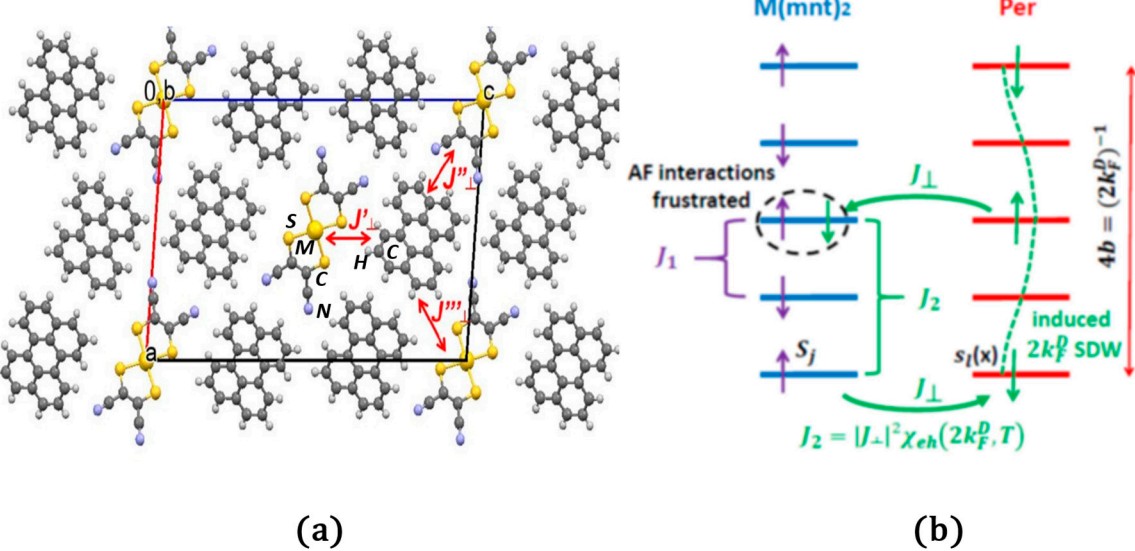

**(a)**                                                                 **(b)**

**Figure 6.** (**a**) Crystal structure of $\alpha$-Per$_2$-M(mnt)$_2$ projected along the stack direction b, which reveals the presence of segregated Per and dithiolate stacks. The atoms are labelled. First, neighbor inter-stack [M(mnt)$_2$]—Per AF exchange coupling $J_\perp$ are schematically indicated (note that there are three different types of interactions per Per). (**b**) Schematic representation of competing S = 1/2 AF exchange couplings in $\alpha$-Per$_2$-M(mnt)$_2$ derivatives for M = Ni, Pd and Pt. $J_1$ is the first neighbor direct AF exchange interaction on dithiolate stack. $J_2$ is the second neighbor indirect RKKY exchange interaction mediated by the induced $2k_F^D$ SDW on the Per stack.

Here, we consider specifically the Pd salt, where the effects of SP instability and of Kondo coupling are very remarkable. Figure 7 shows the thermal dependance of the spin susceptibility of the Pd salt obtained from ref. [36]. At high temperature, $\chi_{spin}$ follows the thermal dependance of the spin susceptibility of the S = 1/2 AF Heisenberg Hamiltonian (1) with J = 270 K ($J_1$ in Figure 6b). Then, below ~100 K $\chi_{spin}$ drops strongly when 1D SP structural correlations on the Pd(mnt)$_2$ stack develop [37] (see insert of Figure 7). This behavior bears some resemblance to that exhibited by (BCP-TTF)$_2$AsF$_6$ below $T_{SP}^{MF}$ (Figure 4), with, however, a larger rate of decrease of $\chi_{spin}$. The rate of decrease is enhanced below about 40 K, as revealed by a change of slope (brown dotted lines in Figure 7). This behavior is ascribed to the relevance of the Per-dithiolate exchange Kondo interaction, which sets a $J_2$ AF interaction of the same sign as $J_1$ (see Figure 6b and ref [25] for more detail). This induces a drop of $\chi_{spin}$ which is so rapid that, at the SP transition of 28 K, only 1/3 of $\chi_{spin}$ remains (this is different to (BCP-TTF)$_2$AsF$_6$, where, without AF frustration, 90% of $\chi_{spin}$ remains at $T_{SP}$—see Figure 4).

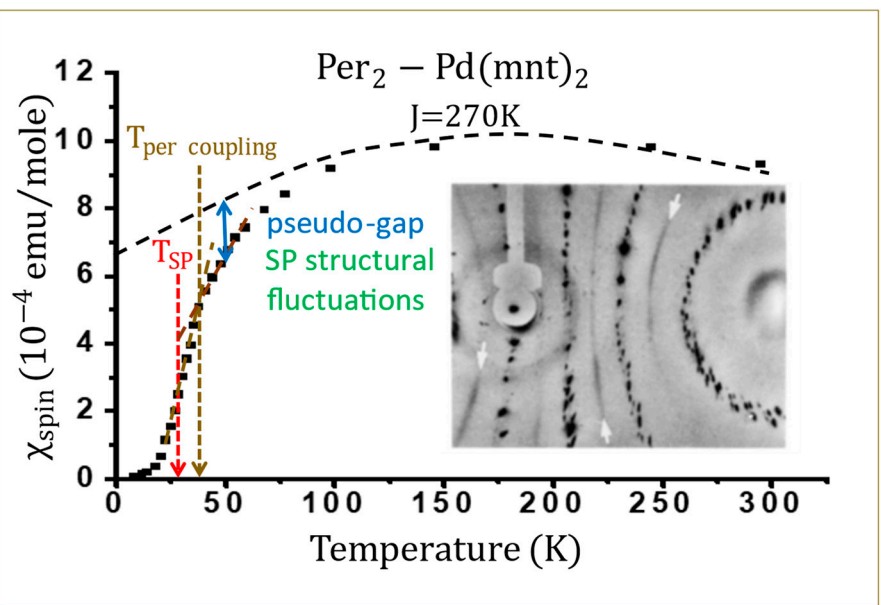

**Figure 7.** Thermal dependence of the spin susceptibility of $Per_2$-Pd(mnt)$_2$ (data taken from ref. [36] and kindly provided by M. Almeida). At high temperature, $\chi_{spin}$ behaves as the spin susceptibility of the S-1/2 AF Heisenberg Hamiltonian (1) with J = 270 K on the dithiolate stack. Below 100 K, $\chi_{spin}$ deviates from this dependence due to the onset of SP fluctuations forming a pseudo-gap. Below 40 K ($T_{per\ coupling}$), the deepest rate of decrease is due to the Kondo coupling with the Per stack. The X-ray pattern in inset reveals the presence of 1D-SP structural fluctuations on the Pd(mnt)$_2$ stack at 40 K which are responsible for the pseudo-gap regime (see [37] for more details). The 3D SP transition achieving the condensation of diffuse lines into superstructure spots occurs at $T_{SP}$ = 28 K.

## 6. Spin-Ladder Behavior in the (DT-TTF)$_2$M(mnt)$_2$ Series

Spin-ladders are a class of low dimensional magnetic system built with a finite number of magnetically (generally AF) coupled spins chains. These spin systems are intermediate between 1D isolated magnetic chains and the 2D magnetic layer. However, their unexpected magnetic properties have attracted a lot of attention relevant for the study of low dimensional and topological quantum systems [38]. Earlier studies show that, depending on the number of interacting spin chains (legs in the spin-ladder), quite different magnetic behaviors are observed. AF coupled S = 1/2 spin-ladders with an odd number of legs behave as isolated chains with a finite spin susceptibility upon approaching zero temperature and spin–spin correlations with a power-law dependance. This contrasts with spin-ladders with an even number of legs, which present a gap in the magnetic excitation spectrum so that the spin susceptibility drops exponentially towards zero upon cooling into a spin-liquid ground state lacking long-range order. The even-leg ladders can be seen as spin singlet pairs, with spin–spin correlation decaying exponentially due to the presence of a finite spin gap.

Two-leg spin ladders are found both in inorganic and organic states. In the inorganics, such as the transition metal oxide $\alpha'$-NaV$_2$O$_5$ and the family of 1D incommensurate composite crystals M$_{14}$Cu$_{24}$O$_{41}$ with M = La, Y, Sr, Ca, ladders exhibit a rectangular structure with transverse AF exchange interactions along the rung of the ladder. In the organics considered below, the two legs are linked by AF zig-zag interactions.

Organic ladders were first reported in the DTTTF$_2$-M(mnt)$_2$ family of 2:1 charge transfer salt whose structure is represented in Figure 8. These salts are composed of layers of alternating dithiolate stacks and double stacks of DTTTF donors. Double stacks are related by screw axis symmetry, so that donors form a zig-zag chain. (DT-TTF)$_2$M(mnt)$_2$ salts with M = Au, Cu, Pt and Ni are the first organic materials found to exhibit spin-ladder physics [39]. Since then, many other organic compounds, reviewed in [40], have exhibited similar spin-ladder behavior.

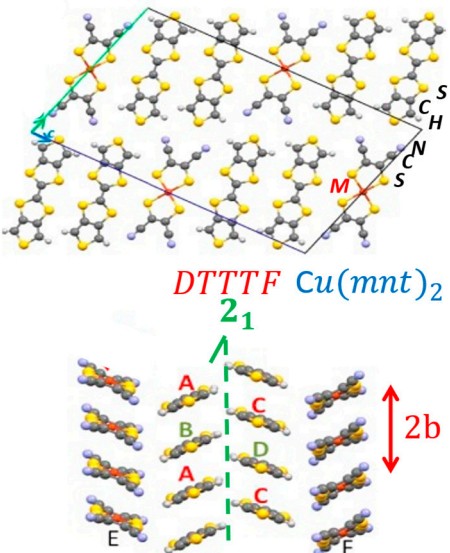

**Figure 8.** Crystal structure of DTTTF$_2$-Cu(mnt)$_2$ perpendicular to the stack direction (top) and along the stack b direction (bottom), showing the 2$_1$-screw axis symmetry between donor stacks in the metallic state. The atoms are labelled in the top figure. The break of 2$_1$ symmetry below the 4k$_F$ transition doubles the b periodicity and leads to a charge disproportion with alternating hole rich (red) and hole poor (grey) donors.

The most interesting compound, DTTTF$_2$-Cu(mnt)$_2$, incorporates the non-magnetic close shell Cu(mnt)$_2$ dithiolate molecule. It is a quarter-filled D$_2$X 1D metal at ambient conditions with the P2$_1$/c monoclinic space group. It undergoes a second-order electronic transition at 235 K ($T_{4k_F}$) [41], from a weakly localized regime to gapped insulator, together with the development of a 3D superstructure doubling the b periodicity [41] (Figure 9a). As this transition leaves the spin susceptibility unaffected [42] (Figure 9b), only the charge degrees of freedom are involved. The metal insulator transition thus corresponds, for a 1D system, to a 4k$_F$ charge localization (Figure 2). One should discard a 2k$_F$ Peierls transition, which should also open a (non-observed) gap in the spin degrees of freedom. In a single chain compound, a 4k$_F$ charge localization occurs either on one bond out of two (4k$_F$ BOW) or one site out of two (4k$_F$ CDW). These two types of 4k$_F$ modulation exhibit an inversion symmetry either on the bonds or on the sites, respectively. In the case of the DTTTF zig zag ladder, where the 2$_1$-screw axis is removed (Figure 8) the inversion symmetry elements located on the left and right chains forming the ladder in its metallic state are both removed. Thus, the 4k$_F$ modulation appears to be a mixture of BOW and CDW (or CO) modulations (see Section 7). However, the structural refinement performed at 120K in the Cu derivative [43] shows, basically, the formation of CDW stack alternating "neutral" and "ionic" nearly equidistant donors along b (lower part of Figure 8). The same type of 4k$_F$ transition was previously reported in DTTTF$_2$-Au(mnt)$_2$ [39], with, however, the observation of a short-range order below ~220 K instead of the long-range order below 235 K in the Cu salt. In the Au derivative, infra-red spectroscopy indicates an incomplete charge disproportion (0.5 ± δ) of about 2δ~0.35–0.4 between donors [44]. Finally, note that inversion symmetry, located in the middle of the ladder, and relating its two peripheric chains (see Figure 11 at the end of the section), could be absent since the structural refinement of the Cu derivative [43] indicates that, with different charges, the inversion symmetry is weakly broken between stacks (lower part of Figure 8).

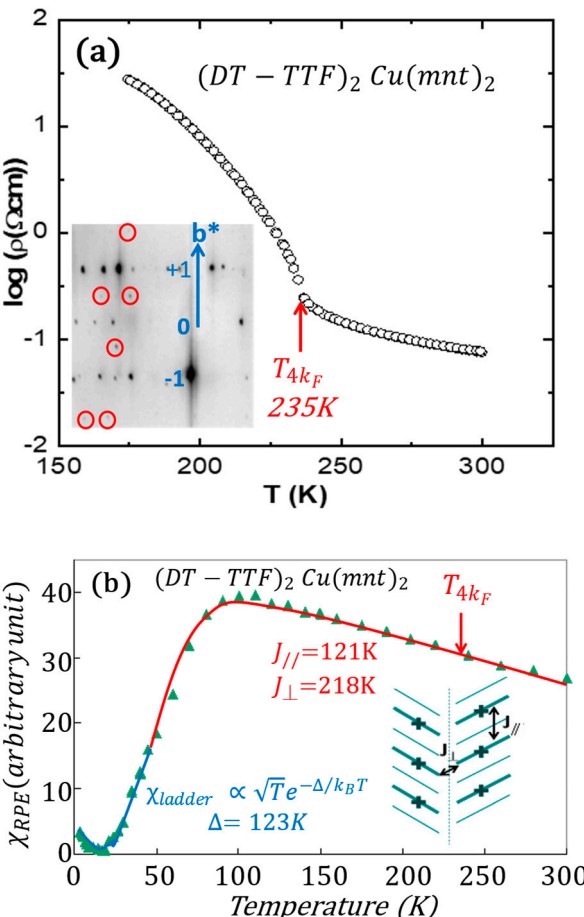

**Figure 9.** Thermal dependance of the logarithm of the resistivity (**a**) and of the RPE spin susceptibility (**b**) of DTTTF$_2$-Cu(mnt)$_2$ (data originally published in ref. [41,42], respectively, are kindly provided in a modified form by M. Almeida). The b*/2 superlattice reflections appearing below $T_{4k_F}$ are shown (red circles) in the X-ray pattern of the insert of figure (**a**). The thermal dependance of $\chi_{RPE}$ if fitted at high temperature by $\chi_{spin}$ of the frustrated ladder (red curve) with $J_\perp$ and $J_{//}$ is represented in the figure, and at low temperature by $\chi_{ladder}$ (blue curve), whose analytical expression is given. In (**a**), the weak increase of resistivity on approaching $T_{4k_F}$ from higher T could be due to the development of a pseudo-gap, due to 4k$_F$ fluctuations in the metallic state.

The most interesting aspect of DTTTF$_2$-Cu(mnt)$_2$ concerns the thermal behavior of the spin susceptibility, which reveals the formation of a spin gap, $\Delta$. Above 50 K, the fit of $\chi_{spin}$ allows the determination of frustrated AF exchange interactions; they are nearly twice as large along the zig-zag (rung) direction ($J_\perp$ = 218 K) as along the stack direction ($J_{//}$ = 121 K)—see the insert of Figure 9b [42]. In this temperature range, the spin gap result from frustrated AF interactions between $J_\perp$ and $J_{//}$. In this respect, the zig-zag ladder chain bears some resemblance with the $J_1$-$J_2$ frustrated AF chain on the Pd(mnt)$_2$ dithiolate stack considered in Section 5. The spin gap is more visible at a low temperature, where the spin susceptibility exhibits activated behavior below 50 K (Figure 9b). Here, $\chi_{spin}$ clearly behaves as the spin susceptibility of an SP compound (see Section 4). From the high and low temperature fits, the same spin gap value $\Delta \approx 123$ K is obtained [42]. However, since the high temperature and low temperature gap values are nearly identical, there is no apparent magnetic symmetry breaking in the temperature, from which one deduces that there is no real enhancement of a low temperature 2k$_F$-SP lattice modulation. However, optical measurements indicate a modification of the vibrational spectrum below 70 K in the Au derivative [45], which has been attributed to the formation of the spin gap. Therefore, the problem of the coexistence of 2k$_F$ SP and of 4k$_F$ charge modulations in the whole

temperature range remains ambiguous. Below, we provide an argument for the presence of a high temperature SP modulation.

The electronic structure of the zig-zag ladder in its quarter-filled metallic state is quite subtle. A simple tight binding analysis of the ladder structure shown in Figure 10a, which reveals the presence of two bands. Depending on the ratio of zig-zag ($t_d$) and intra-chain ($t_s$) transfer integrals, two different conduction band fillings can be found. In a free electron representation, one obtains the following: (1) for dominant $t_s$ (>1.7$t_d$), two partially-filled 1D conduction bands which could exhibit a combined inter-band 2($k_{F1}$+ $k_{F2}$) nesting process at b*/2 (Figure 10b); and (2) for dominant $t_d$ (>0.58$t_s$), an upper partly filled 1D conduction band with a simple 2$k_F$ nesting process could occur at the same b*/2 wave vector (Figure 10c). A more accurate electronic structure determination, with a dispersion resembling the ones shown in Figure 10, is calculated in ref. [46] with an extended Hückel Hamiltonian. With a $t_s$/$t_d$~1.7 ratio, the electronic structure of the Au derivative is at the borderline between dispersions given in Figure 10b,c. However, with a larger $t_s$/$t_d$~3 ratio, the Ni and Pt derivatives have a two-conduction band structure, as shown in Figure 10b.

### Quarter filled zig-zag ladder : ρ=1/2 hole per donor

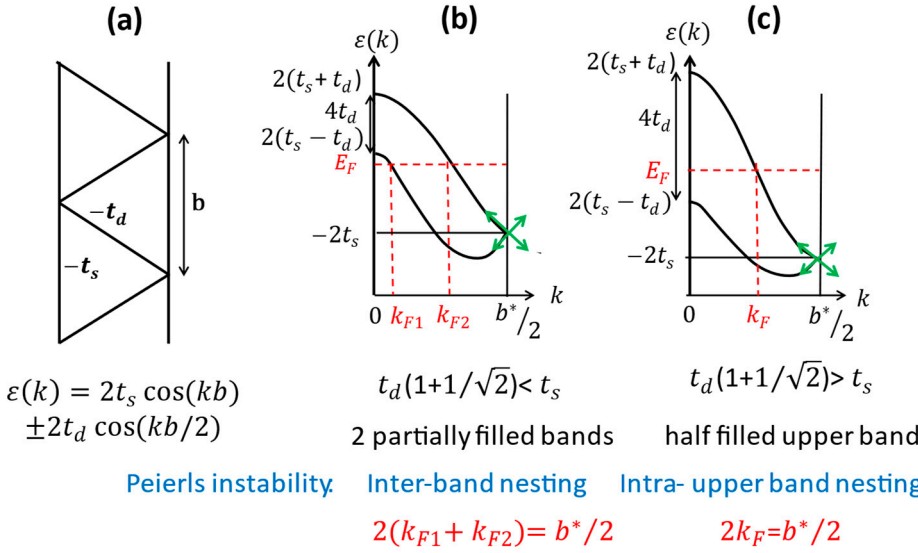

**Figure 10.** Band structure of the quarter fill zig-zag ladder shown in (**a**). With one-half hole per donor and for decreasing ratio of transfer integrals $t_s$/$t_d$ one shifts from a two partially filled band structure (**b**) to a half-filled upper band structure (**c**). The Peierls instability is achieved by an inter-band nesting process in (**b**) and by an intra-upper band nesting process in (**c**). For half a hole per donor, both FS nesting processes always occur at b*/2.

At first glance, the stabilization of a 2$k_F$ Peierls ground state, as predicted by the free electron band structures of Figure 10, does not correspond to physical reality because $\chi_{spin}$ does not drop at $T_{4k_F}$ (Figure 9b). However, a 2$k_F$ SP instability of the zig-zag lattice of localized spin, which exhibits a modulation of the same symmetry as a 2$k_F$ Peierls instability on the same zig-zag chain, could be stabilized without affecting $\chi_{spin}$. Here, the quadrupling of the zig-zag periodicity d in a diagonal direction (2$k_F^d$ = d*/4) due to an SP instability induces a doubling of the chain periodicity along b, as does the 4$k_F^b$ = b*/2 charge localization in the stack direction. Thus, as shown in Figure 11, both types of instabilities of the same periodicity could coexist. This argument is assessed from theoretical calculations of the instability of a quarter-filled ladder chain treated with the extended Hubbard Hamiltonian coupled to a phonon field [47]. The CDW and BOW patterns thus calculated, assuming an important diagonal coupling, are shown in Figure 11. In this representation it is easy to understand that the spin gap should open from spin singlet pairing along the diagonal bond exhibiting the strongest AF exchange coupling,

$J_\perp > J_{//}$. However, this interpretation should be validated by a precise determination of the SP modulated structure.

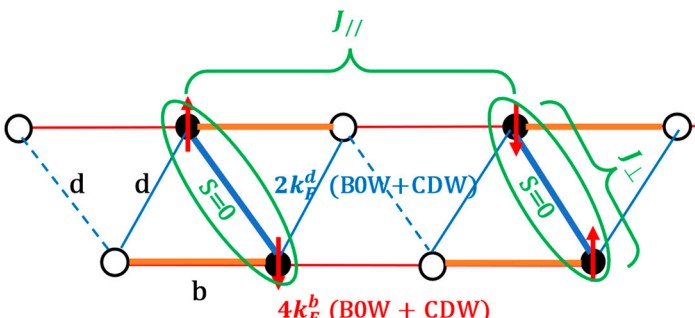

**Figure 11.** Schematic structure of the BOW and CDW in the quarter-filled zig-zag ladder combining $2k_F^d$ modulation in zig-zag (blue) and $4k_F^b$ modulation in stack (red) directions. Frustrated AF exchange interactions are indicated, as well as the location (in green) of spin singlets. Note that, in this figure, the centrosymmetric bond and charge patterns are kept for simplicity.

More complex behaviors, not yet completely understood, are found in the DTTTF$_2$-M(mnt)$_2$ series with M = Pt and Ni [46,48]. As for the Per$_2$-M(mnt)$_2$ materials with the same M (Section 5), the complexity arises from the additional presence of dithiolate paramagnetic ions forming an S = 1/2 AF chain, which compete with the frustrated AF interactions previously found in the donor ladder. DTTTF$_2$-M(mnt)$_2$ with M = Pt and Ni is a better conductor than the Au and Cu analogues. However, they still exhibit a metal–insulator transition, but at a twice smaller $T_{4k_F}$ [46,48]. In the Pt derivative the transition stabilizes the 2b periodicity as for the Cu and Au derivatives [46]. However, the Pt and Ni salts exhibit two interacting magnetic subsystems, as shown by EPR measurements for which the width of the single line is observed to increase dramatically as the conductivity increases. Both salts show two different types of AF interaction, and at a lower temperature exhibit a transition not yet fully understood: it is possibly of the AF type in the Ni salt and of the SP type in the Pt salt. The SP instability of the ladder found in the Au and Cu derivatives is apparently less stable in Pt and Ni derivatives. However, it could be restored by the dimerization of the dithiolate stack, setting a 2b SP lattice periodicity, as for Per$_2$-M(mnt)$_2$. However, the Pt derivative exhibits upon cooling an enhanced disorder between neighboring stacks, which is certainly caused by frustrated coupling between dimerized dithiolate and DTTTF stacks.

## 7. Coexistence of $4k_F$ BOW and $4k_F$ CDW Orders in Quarter-Filled Systems

The spin ladder DTTTF$_2$-M(mnt)$_2$ with M = Au or Cu superimposes, in a single phase, three modulations which generally occur through different transitions in the Fabre salts (TMTTF)$_2$PF$_6$: namely a $4k_F$ BOW on the dimerized stack in as grown material, which is enhanced below $T_\rho$~230 K, and then a $4k_F$ CDW around $T_{CO}$~60 K followed by a low temperature SP transition at 18 K. The $4k_F$ modulations are interesting because the BOW destroyed the inversion centers on the site and the CDW on the bonds (Figure 2). As a result, the combined $4k_F$ (BOW + CDW) modulation destroys all the inversion centers so that each stack becomes polar. This property is conserved in the 3D stack array of (TMTTF)$_2$X with octahedral anion X, so that these Fabre salts exhibit electronic ferroelectricity below $T_{CO}$ [49]. Note that, in addition, the 3D ferroelectric ground state should be stabilized by a cooperative displacement of the anions X towards the hole rich donors [16,50].

In spite of the presence of giant dielectric anomalies at $T_{CO}$, structural modifications are weakly apparent in the ferroelectric phase. Charge disproportion is revealed, in addition to optical means [51], by X-ray determination of the modulation of electronic density (and the loss of inversion symmetry in the structure) [52]. Atomic displacements below $T_{CO}$ were evidenced by neutron diffraction experiments [53], which probably reveal the deformation of the H-bond network, which couple methyl terminal groups of the TMTTF with the

anions [16]. The $T_{CO}$ transition is also revealed by small lattice anomalies probed by high resolution thermal expansion measurements in the c* direction, where donor layers and anions alternate [54,55]. However, the most surprising finding is the observation of a splitting of the ESR spectrum in the CO ground state, with magnetic axes rotation indicating a broken magnetic symmetry [56–58]. Such measurements reveal, in addition to a modification of the thermal dependence on the ESR linewidth ascribed to an enhanced anion displacement below $T_{CO}$ [57], a superlattice modulation between the spin chains along c [58]. This unexpected result has been interpreted as linked to a space modulation of the CDW, whose origin remains elusive because there is no structural evidence of a doubling of the c periodicity while keeping the ferroelectric phase.

Finally, note that defects of charge ordering, inducing phase shift in the $4k_F$ CDW, strongly perturb the ferroelectric order. This is, in particular, the case for irradiation defects in the Fabre salts [50]. Here, defects nucleate local polarization and induce a relaxor type of dynamics.

## 8. Soliton Nucleation in Perturbed Spin-Peierls Compounds

The low energy magnetic excitations of the Heisenberg chain are a continuum of non-interacting spinons [59,60]. In this continuum, in the presence of a magnetoelastic coupling, "$2k_F$" pre-transitional SP lattice fluctuations progressively develop, in the adiabatic limit, a pseudo-gap below $T_{SP}^{MF}$, as previously discussed in Section 4. The energy dispersion of the pseudo-gap has been measured by inelastic neutron scattering in the Fabre salts [32]. Issued from the magnetic excitation spectrum, a fraction of spinons is pinned by defects under the form of solitons. This is revealed by the detection of Rabi oscillations of pinned solitons in AF magnetic chains of $(TMTTF)_2X$ (X = $PF_6$, $AsF_6$) [61] and o-$DMTTF)_2X$ (X = Cl, Br) [62]. The formation of static solitons breaks the longitudinal coherence of the SP dimerized order, as schematically represented in Figure 12. In this situation, and in order to recover the optimal lateral coupling between dimers located on neighboring SP chains, each defect should nucleate a pair of a soliton and anti-soliton, where each dimerization defect bears an unpaired S = 1/2. Defects thus reduce the intrachain SP correlation length, on $\xi_{//}$, but also break the inter-chain SP phase–phase correlations more drastically (Figure 12). In this situation, the superstructure reflections of the SP ground state are transformed into diffuse lines (the Fourier transform of the local SP order) whose width allows the determination of $\xi_{//}^{-1}$. Diffuse lines thus remain the signature of a local (basically 1D) SP order spatially limited by the presence of defects and various disorders. This effect is well documented in irradiated Fabre salts, where point defects due to irradiation centers break the SP lattice coherence and create unpaired S = 1/2, which manifest with a low temperature Curie-type paramagnetic response revealed by ESR [63,64]. Detailed ESR studies of defects have been performed in as-grown $(TMTTF)_2X$ [61], o-$(DMTTF)_2X$ [62,65,66]. One-dimensional local SP correlations are also revealed in as-grown $Per_2M(mnt)_2$ systems with M = Ni [67] and Pt [25] compounds. Unpaired spins 1/2 have been identified by low temperature NMR measurements in the Pt compound [68]. Conversely, for low amounts of substituent, local 1D–SP structural correlations coexist with 3D–SP dimerization superstructure reflections in Si doped $CuGeO_3$ [69]. This proves that dimerization in SP organics is more sensitive to defects than in SP inorganics such as $CuGeO_3$. The SP pattern of $CuGeO_3$ doped with various elements has been quantitively analyzed in ref. [69].

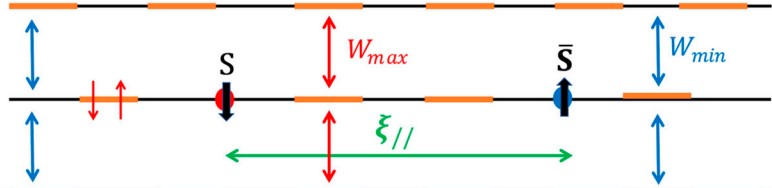

**Figure 12.** Schematic representation of a pair of a spin 1/2 soliton (S) and spin+1/2 anti-soliton ($\bar{S}$) shifting the phase of the SP intra-chain dimerization on $\xi_{//}$. In addition, between the S and $\bar{S}$ entities the phase shift between neighboring dimers changes from $\pi$ to 0. This enhances the inter-dimer coupling energy from $W_{min}$ to $W_{max}$, as schematically indicated.

Note that coherent spin dynamics of solitons have been detected by ESR in the organic spin chain compounds (o-DMTTF)$_2$X [62] and (TMTTF)$_2$X [70] with natural defects. More precisely, defects polarize the AF coupled spins in their vicinity. This leads to a finite local alternating magnetization around the site of the defect, which can be described in terms of a soliton, i.e., a S-1/2 quasiparticle built of many correlated spins, pinned to the defect. Thus, in the presence of a large number of defects, the S-1/2 quasiparticles can interact AF so that the overlap between these magnetic regions is able to achieve a 3D AF order. Of course, this order coexists with local SP correlated 1D regions previously considered. The coexistence of both types of order are observed [71,72] in the quarter-filled organic salt (TMDMTSF)$_2$PF$_6$, where the TMDMTSF non-centrosymmetric donor, built with one half of the TMTTF molecule and one half of the TMTSF, is randomly oriented in a stack direction. Similar phase diagrams revealing a coexistence between SP and AF orders are determined in the SP system CuGeO$_3$ substituted by various elements [73].

Finally, note that under a high magnetic field of the magnitude of the SP gap, the SP dimerized ground state is destabilized towards a lattice of periodic solitons exhibiting an incommensurate periodicity in chain direction, which varies with the amplitude of the applied magnetic field. The resulting phase diagram is observed in many SP compounds [74], including the organic Fabre salts [75] under a high magnetic field. The structure of the high field soliton lattice of CuGeO$_3$ has been determined both by magnetic and structural measurements [76,77].

## 9. Concluding Remarks

The organic conductors where electronic kinetic energy is comparable to Coulomb repulsion provide an exceptional panel of correlated systems in which unexpected ground states are stabilized. These effects are more clearly revealed in systems exhibiting a 1D electronic anisotropy, and in compounds with a commensurate charge transfer such as the quarter-filled "conductors". This leads, in particular, to spin-charge decoupled physics and the stabilization of new ground states such as the spin-Peierls one. Such ground states are especially stabilized in the organic state, presenting an important lattice response due to the presence of an underlying soft molecular lattice, which thus provides a significant magneto-elastic coupling. In this review, we summarize such features in three families of quarter filled donor organics: (1) the single stack D$_2$X salts which provide clear-cut examples of spin-charge decoupling and low temperature SP instabilities; (2) the segregated two-stack Per$_2$-M(mnt)$_2$ compounds where the Kondo coupling between the AF dithiolate chain and the metallic Per chain unexpectedly enhance the SP instability; and (3) the spin ladder DTTTF$_2$-M(mnt)$_2$ compounds where the double donor stack forming a zig-zag ladder exhibits, simultaneously, a charge localization and an SP gapped ground state. In this review, we have especially presented a unified description of the electronic and structural instabilities in these families of materials.

**Funding:** This research received no external funding.

**Institutional Review Board Statement:** Not applicable.

**Informed Consent Statement:** Not applicable.

**Data Availability Statement:** Not applicable.

**Acknowledgments:** This review is a tribute to Manuel Almeida for his 70th birthday. Its content is in great part constructed on scientific results obtained in collaboration with him and his research team at the Universidade de Lisboa. A collaboration between Lisboa and the Laboratoire de Physique des Solides (LPS) of Université Paris-Sud was initiated at the beginning of 1980s by Luis Alcácer, then by Manuel Almeida with long stays at LPS of R.T. Henriques and V. Gama during their PhD period in my group and the one of D. Jérome. With time, the initial research on Per$_2$-M(mnt)$_2$ was extended to the study of the parented spin ladder organics DTTTF$_2$-M(mnt)$_2$ in collaboration with C. Rovira in Barcelona and M. Almeida in Lisboa. The other half of the review is devoted to more classical SP organics, with a special emphasis on recent results obtained in Fabre and analogous salts incorporating substituted donors in collaboration with M. Fourmigué, P. Foury-Leylekian and S. Petit. For this review, fruitful discussions with P. Alemany, S. Bertaina, C. Bourbonnais and E. Canadell are acknowledged.

**Conflicts of Interest:** The author declares no conflict of interest.

### Annex: Chemical Names of Organic Molecules Quoted in the Main Text

| | |
|---|---|
| Per | perylene |
| (CH)$_x$ | polyacetylene |
| TCNQ | tetracyanoquinodimethane |
| Qn | quinolinium |
| NMP | N-methylphenazinium |
| MEM | N-methyl-N-ethylmorpholinium |
| TTF | tetrathiafulvalene |
| DIMET | dimethyl-ethylene-tetrathiafulvalene |
| t-TTF | trimethylene-tetrathiafulvalene |
| TMTTF | tetramethyl-tetrathiafulvalene |
| TMTSF | tetramethyl-tetraselenafulvalene |
| BCP-TTF | benzo-cyclopentyl-tetrathiafulvalene |
| o-DMTTF | *ortho*-dimethyl-tetrathiafuvalene |
| DTTTF | dithiophen-tetrathiafulvalene |
| mnt | maleonotril-dithiolate |

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
