# Peer review of "Spin-Peierls, Spin-Ladder and Kondo Coupling in Weakly Localized Quasi-1D Molecular Systems: An Overview"

_magnetochemistry, doi:10.3390/magnetochemistry9020057_

Round 1

Reviewer 1 Report

This short review article deals with the magneto-structural properties of electron-electron correlated quasi-1D molecular organics with that constitute weakly localized quarter-filled metallic-like systems with important spatial anisotropy in the antiferromagnetic interactions due to the stacking structure. These systems are remarkable because electronic kinetic energy is similar in strength to the Coulomb repulsion, providing a large range of unconventional properties and ground states.

The paper is organized in several short sections illustrating specific situations or describing some specific systems. After a short introduction to the basic interactions in quasi-1D molecular systems and the generic description of this family of system, the author describes through examples: spin-Peierls fluctuations and gap formation, Kondo coupling between localized and delocalized stacks, spin-ladder, coexistence of bond-ordered wave (BOW) and charge-density wave (CDW) orders, and soliton nucleation in perturbed spin-Peierls compounds. In these examples, the author presents recent example and some helpful state-of-the-art information. The subtle balance between charge and spin degrees of freedom, structural peculiarities (“soft molecular lattice”) and the effective dimensionality of the magnetic Hamiltonian is a key challenge to understand the observed properties. This is nicely done in a concise way by the author.

On a more general tone, the paper is well organized in the spirit of a short review. The data and the concepts are clearly presented and the bibliographical information is up to the standard. My recommendation is thus to accept the paper for publication.

Author Response

I warmly thank the reviewer for his (her) positive comment concerning the general interest and detailed content of this mini-review for the literature. 

As report 1 does not adress precise comment on specific detail of the text, the manuscript has not been changed according to the content of the report. Nevertheless, the manuscript has been improved according to comment of  others referees, and correction of mistake found by the autor.   

Reviewer 2 Report

The work is devoted to analytical review of magneto-structural properties of strongly correlated quasi-one-dimensional organic materials like charge-transfer stacked crystals. These materials may have special types of the ground states like spin-Peierls state and peculiarities in the lowest excitation energy spectra due to relatively strong electron-electron repulsion and significant magneto-elastic coupling. Three families of so-called quarter filled donor organics were considered in details from this common point of view.

To my mind, the above review can stimulate new theoretical and experimental researches in field of low-dimensional organic conductors and nanomagnets. It will be useful definitely for the targeted design of new materials for nanoelectronics and spintronics. The manuscript under review can be published without corrections.

Author Response

I warmly thank the reviewer 2 for his (her) positive comment concerning the general interest and detailed content of this mini-review for the literature. 

As report 2 does not address precise comment on specific detail of the text, the manuscript has not been changed at this level. Nevertheless, the manuscript has been improved according to comment of others referees, and correction of mistake found by the author.  

Reviewer 3 Report

The manuscript by J.-P. Pouget is a comprehensive review of physical properties and exotic ground states of organic antiferromagnetic systems with the reduced dimensionality. Certainly, this manuscript will be interesting for the readers of Magnetochemistry after minor improvements:

1) Abbreviations in the Abstract and throughout the text should be explained. It is highly challenging and not reader friendly to go through abbreviations of organic compounds without explanations. The complete list of abbreviations (organic compounds and physical terms) should be given as a separate section of this manuscript.

2) Captions of Figures 1 and 2: “is in heavy” should be “written in bold text”.

3) Figures 6 and 8: the color scheme should be explained, describing the atoms presented.

Author Response

I thank referee 3 for his (her) useful comments.

We have taken into account all the remarks done by the referee.

1- I have replaced some abreviation in by the full name in order to render the text more readable.

2- I have introduced in the annex the extended name of organic molecules quoted in the text 

3- I have included in figures 6 and 8 the atomic symbol of the colored atoms 

4- I have corrected the captions of figures 1 and 2.

Reviewer 4 Report

The manuscript contains a review on quasi-one-dimensional molecular conductors, which are correlated systems characterized by the presence of magnetoelastic interactions between spins and a relatively soft underlying lattice. This leads to the stabilization of the unique ground states in these materials. The paper gives an extended overview of selected families of materials and their characteristic ground states resulting from magnetoelastic interactions, such as spin-Peierls states, Kondo coupling, but also Rabi oscillations, which are important from the point of view of quantum computing.

The article is well organized and provides 75 references, which are important both from the point of view of the historical development of the research area and the current achievements. I recommend the manuscript for publication in Magnetochemistry after minor revisions described below.

1. I recommend that abbreviations be used with care, as too many abbreviations inevitably lead to difficulties in understanding the text. In particular, I encourage the author to use 'antiferromagnetic' instead of 'AF', 'donor' instead of 'D', etc.

2. Please correct the reference numbering in lines 337 (Ref. 44 instead of 45) and 351 (Ref. 45 instead of 44).

Author Response

I thank the referee for his (her) positive report pointing the interest of my minireview for the literature.

I have followed the referee suggestion to suppress many abbreviations in order to render the text more readable.

I thank also the referee for pointing out errors in the reference numbering. I fact after a new reading of the text I have found and corrected other errors.
